# The Functional Interaction of EGFR with AT1R or TP in Primary Vascular Smooth Muscle Cells Triggers a Synergistic Regulation of Gene Expression

**DOI:** 10.3390/cells11121936

**Published:** 2022-06-16

**Authors:** Virginie Dubourg, Barbara Schreier, Gerald Schwerdt, Sindy Rabe, Ralf A. Benndorf, Michael Gekle

**Affiliations:** 1Julius-Bernstein-Institute of Physiology, Martin Luther University Halle-Wittenberg, 06112 Halle, Germany; barbara.schreier@medizin.uni-halle.de (B.S.); gerald.schwerdt@medizin.uni-halle.de (G.S.); sindy.rabe@medizin.uni-halle.de (S.R.); michael.gekle@medizin.uni-halle.de (M.G.); 2Department of Clinical Pharmacy and Pharmacotherapy, Institute of Pharmacy, Martin Luther University Halle-Wittenberg, 06120 Halle, Germany; ralf.benndorf@pharmazie.uni-halle.de

**Keywords:** EGFR, AT1R, TP, synergism, transcriptomic

## Abstract

In vivo, cells are simultaneously exposed to multiple stimuli whose effects are difficult to distinguish. Therefore, they are often investigated in experimental cell culture conditions where stimuli are applied separately. However, it cannot be presumed that their individual effects simply add up. As a proof-of-principle to address the relevance of transcriptional signaling synergy, we investigated the interplay of the Epidermal Growth Factor Receptor (EGFR) with the Angiotensin-II (AT1R) or the Thromboxane-A2 (TP) receptors in murine primary aortic vascular smooth muscle cells. Transcriptome analysis revealed that EGFR-AT1R or EGFR-TP simultaneous activations led to different patterns of regulated genes compared to individual receptor activations (qualitative synergy). Combined EGFR-TP activation also caused a variation of amplitude regulation for a defined set of genes (quantitative synergy), including vascular injury-relevant ones (*Klf15* and *Spp1*). Moreover, Gene Ontology enrichment suggested that EGFR and TP-induced gene expression changes altered processes critical for vascular integrity, such as cell cycle and senescence. These bioinformatics predictions regarding the functional relevance of signaling synergy were experimentally confirmed. Therefore, by showing that the activation of more than one receptor can trigger a synergistic regulation of gene expression, our results epitomize the necessity to perform comprehensive network investigations, as the study of individual receptors may not be sufficient to understand their physiological or pathological impact.

## 1. Introduction

The cellular changes occurring upon the activation of cell surface receptors have mostly been studied in a linear manner, meaning that only one receptor has been considered at a time, often using targeted approaches. However, the in vivo reality consists of a complex situation with multiple extracellular influences. This results in the simultaneous activation of various receptors that are connected to intertwined downstream signaling pathways with different outcomes.

We investigated the relevance of considering these aspects in the study design to draw meaningful conclusions about the full impact of a receptor on cell function. To do so and in order to capture a global and advanced picture of the complex intracellular system mobilized, we analyzed the effect of individual or combined activation of various receptors at the transcriptomic level. This strategy allowed for the observation of the crosstalk in between receptors and the consequences on gene expression regulation: the simultaneous activation of two receptors can theoretically trigger either an additive response (the effect of the simultaneous activation is equal to the sum of the effect of each receptor) or a synergistic response (the effect of the simultaneous activation is different to the sum of the effect of each receptor). For the latter, qualitative (different pattern of regulated genes) and quantitative (different amplitude of regulation for a defined set of genes) responses can be distinguished.

We chose to work on murine primary aortic vascular smooth muscle cells (aVSMC), as the prominent role of this cell type and of this vascular segment in cardiovascular diseases, one of the main causes of death worldwide [1], makes them a pertinent study case. Moreover, by working on primary cells in culture, we opted for the ultimate strategy to get around the lack of suitable immortalized cell lines for this model while having a system in which each parameter can tightly be controlled, which could not be achieved in vivo.

Furthermore, we focused on three receptors known to be involved in cardiovascular system physiological functions (e.g., vascular tone, blood pressure regulation [2,3,4]) and pathophysiological conditions (e.g., vascular remodeling, hypertension, chronic cardiovascular diseases [5,6,7]), and to functionally interact: the Epidermal Growth Factor (EGF) receptor (EGFR), the Angiotensin II (AngII) type 1 receptor (AT1R) and the thromboxane A2 receptor (TP). The former is a tyrosine kinase receptor that can be activated by ligand-binding or by crosstalk with activated G protein-coupled receptors (GPCR), such as AT1R and TP, through a mechanism called transactivation [8,9,10,11]. EGFR transactivation appears as a critical actor in cardiovascular disease pathophysiology [12], and in vivo and in vitro evidence concerning the role of the crosstalk of EGFR with TP and AT1R in these processes has been piling up [13,14,15].

The aim of this case study was, therefore, to identify the impact of the long-term activation of EGFR and of its crosstalk with AT1R or TP on gene expression in an unbiased approach. Additionally, we intended to generate hypotheses concerning the mechanisms that may lead to these changes as well as their cell biological consequences. To do so, aVSMC were incubated with EGF and/or AngII and/or U46619 (TP agonist), and RNA-sequencing was used to get a snapshot of the transcriptome after 24 and 48 h stimulation. Bioinformatics and differential expression analyses were applied to identify genes regulated following the activation of the various receptors. Furthermore, functional analyses were performed to identify putative regulators and cellular pathways that may link gene expression alteration and the ultimate phenotypical changes previously observed in vivo. Finally, a targeted approach was used to identify genes displaying quantitative synergistic regulation and potentially involved in these phenotypical changes.

We observed that exclusive EGFR and TP-activation led to gene expression regulation, while the activation of AT1R alone did not. Nevertheless, the simultaneous activation of EGFR and AT1R or TP both led to a qualitative synergistic regulation of the transcriptome. Simultaneous EGFR and TP-activation also led to a quantitative synergistic response, including for cardiovascular-relevant genes, such as Kruppel-like factor 15 (*Klf15*) and Osteopontin (OPN, encoded by the gene *Spp1*). Finally, the regulated genes were enriched in cell cycle, and cell senescence-related pathways and the alterations of these processes by EGF and U46619 were confirmed experimentally while bringing to light a putative compensatory effect of EGFR and TP activation. Taken together, our results highlight the necessity to perform comprehensive investigations and to consider several receptors simultaneously since their individual study appears to not be sufficient to understand their physiological or pathological impact.

## 2. Materials and Methods

### 2.1. Buffers, Chemicals, Antibodies and Primers

The buffer compositions, antibodies references and working concentrations, primer sequences and chemical references and providers are listed in the Appendix A.

### 2.2. aVSMC Isolation from Mice

C57BL/6 mice were bred in the facilities of the University of Halle-Wittenberg (Germany). aVSMC were isolated as previously described by Ray et al. [16]. The VSMC purity achieved by our protocol was previously tested [17].

### 2.3. Cell Culture

aVSMC were cultivated in DMEM media (low-glucose media powder diluted in H_2_O, with 2 g/L NaHCO_3_, pH 7.4) supplemented with 10% FCS, 0.14 IU insulin, 10 µg/L EGF, 5 mg/L transferrin, 100 nM hydrocortisone and 30 nM sodium selenite. Before all experiments, the cells were synchronized by incubation in serum and supplement-free DMEM media for 24 h. The latter was also used for further incubations with 10µg/L EGF and/or 10 nM AngII and/or 1µM U46619. All experiments were performed on cells that underwent up to 7 passages. Each replicate corresponds to aVSMC isolated from a different mouse.

### 2.4. RNA Sample Preparation

Total RNA was isolated with BlueZol Reagent as described in the user manual from cells incubated for 24 or 48 h with stimuli. Eventual genomic DNA contaminations were removed with “Turbo DNAse-free kit”, following the “rigorous DNAse treatment” protocol from the manufacturer. All samples were cleaned by ethanol precipitation (with 3M sodium acetate, glycogen and 100% ethanol). The quality of the to-be-sequenced RNA samples was assessed using a 2100 Bioanalyzer (Agilent Technologies, Karlsruhe, Germany). All samples had an RNA Integrity Number (RIN) above 7 (with 10 as the maximal possible value).

### 2.5. RNA Sequencing

The samples to be sequenced were prepared in three batches: (1) samples collected after 24 h incubation with EGF and/or Ang and/or U46619 (*n* = 6), (2) after 48 h incubation with EGF and/or AngII (*n* = 5) and (3) after 48 h incubation with EGF and/or U46619 (*n* = 5). Novogene Co., Ltd. (Cambridge, UK) carried out the sequencing libraries preparation (poly(A) enrichment), and the paired-end sequencing (2 × 150 bp) runs on a NovaSeq6000 Illumina system. Adaptor clipping and data quality control were provided by the service company as well. 

Read mapping to the mouse genome mm10 was performed with HISAT2 (v. 2.1.0) [18], and featureCounts (2.0.0, –M –t exon) [19] was used to count the mapped reads. Gene annotation was performed using BiomaRt (v.2.44.4) [20] to access the Ensembl archive v101. 

### 2.6. Differential Expression Analysis and Functional Analysis

Because the differences in the numbers of raw counts per sample in the two datasets would have led to the biased estimation of regulated genes (Appendix A), differential expression analysis was performed in parallel for the 24-h incubation (sequencing batch 1) and the 48-h incubation (batches 2 and 3 together) data, using edgeR (3.30.3) [21] and DESeq2 (1.28.1) [22]. The multi-variable design—animal + treatment—was used for both tools and analyses to overcome the high variations between the individuals from which aVSMC were isolated (Appendix A). Genes with sufficient counts to be considered in the statistical analyses were filtered using the filterByExpr edgeR function and the independent filtering parameter (α = 0.05) of the DESeq2 results function. Normalization factors were calculated with the “trimmed mean of M value” (TMM) method in the edgeR analysis. Significantly “differentially expressed genes” (DEG) were defined as genes with a false discover rate (FDR) below 0.05 in both DESeq2 and edgeR outputs (overlap of the respective results) and with at least 5 FPM (Fragments per million) on average in one of the sample groups considered for a given comparison (Appendix A). Moreover, genes that were not considered for both analysis rounds were filtered out. “Regularized log transformation” (rlog DESeq2 function) was applied upstream of gene expression clustering. UpSetR (1.4.0) [23] was used as an alternative to Venn diagrams to determine the overlap of DEG lists when more than two lists were considered.

Additionally, to the classical differential expression analysis approach used to identify DEG, we calculated fold changes (FC) based on FPM (calculated with edgeR) to highlight genes that were subject to quantitative synergism. To do so, the FPM-based FC of each DEG were calculated for each biological replicate and for each condition (e.g., FC_Replicate1-EGF_ = FPM_Replicate1-EGF/_FPM_Replicate1-Control_). FCs below 1 (down-regulated genes) were transformed with FC_final_ = −1/FC in order to have symmetrical and zero-center values. The FC thus obtained for multiple incubations were considered as the “measured FC”. Meanwhile, the “expected FC” after double incubations were calculated as FC_expected-Replicate1-Substance1+Substance2_ = FC_Replicate1-Substance1 + FCReplicate1-Substance2_ − 1 (if both FC > 0) and FC_expected-Replicate1-Substance1+Substance2_ = FC_Replicate1-Substance1_ + FC_Replicate1-Substance2_ + 1 (if at least one FC < 0). Paired *t*-test was performed to identify genes whose measured and expected FC were significantly different (*p* < 0.05). We additionally made the distinction in between stimulatory (FC_expected_ < FC_measured_) and inhibitory (FC_expected_ > FC_measured_ or with different signs) quantitative synergistic responses. 

Gene ontology (GO) term enrichment analysis was performed with gprofiler2 (0.2.0 – adapted parameters: correction_method = “false_discovery_rate”, exclude_iea = T) [24] using either all or only down or up-regulated genes. GO terms were defined as significantly enriched if the adjusted *p*-value ≤ 0.001 and enrichment *E* ≥ 3, with *E* = (intersection size/query size)/(term size/effective domain size). Rrvgo (1.0.2) [25] was used to summarize and reduce the significantly enriched GO terms lists (threshold: 0.5). 

Ingenuity Pathway Analysis (IPA) software (Qiagen) was used for Upstream Regulator Analysis on lists of genes significantly regulated [26]. The Ensembl identifiers of the latter were mapped to networks incorporated into the software database. The featured “Comparison Analysis” tool was used to match the different results. Results were filtered for |Z-score| ≥ 2 and adjusted (Benjamini–Hochberg) *p*-value ≤ 0.001. 

### 2.7. Quantitative PCR and Digital Droplet PCR

Reverse transcription (RT) reaction was performed on independent RNA samples (not used for RNA-sequencing and from cells isolated from different mice) with random primers and SuperScript II reverse transcriptase, following the manufacturer’s instructions. qPCR was performed with PlatinumTM SYBRTM Green on a 7900 HT Fast Real-Time PCR System (Applied Biosystems). The 2^−ΔΔ^Ct method, using *Gapdh* and *Eef2* as references for normalization, was used for the relative quantification of genes of interest. The instructions provided in the “QX200 ddPCR EvaGreen Supermix” kit were followed for the absolute quantification of genes of interest by ddPCR.

### 2.8. Western-Blot

Cells were incubated with 10µg/L EGF and/or 10 nM AngII and/or 1µM U46619 or 1µM PMA in HEPES-Ringer buffer for 10 min, 24 h or 48 h at 37 °C. Cells were washed with 1xPBS, lysed with RIPA buffer and sonicated 3 × 10 times. Protein concentrations were determined by BCA assay. For each sample, 50µg of proteins were denatured with 6x Laemmli Buffer for 30 min at 37 °C. Proteins were separated by 10% SDS-PAGE and transferred onto 45µm nitrocellulose membranes. Free binding sites of the membrane were blocked with a 5% solution of non-fat dry milk in TBS-Tween. The membranes were incubated overnight with primary antibodies diluted in 5% BSA in TBS-Tween. EGFR, Phosphorylated EGFR (Y1068), ERK1/2, phosphorylated ERK1/2, GAPDH and HSP90 were detected using IRDye-couple fluorescent secondary antibodies (diluted in 5% solution of non-fat dry milk in TBS-Tween) and an Odyssey imaging system (LI-COR Biosciences, Bad Homburg, Germany). Densitometry analysis was performed with Quantity One software (version 4.6.9, BioRad, Munich, Germany). GAPDH and HSP90 were used as references for relative quantification.

### 2.9. Caspase-3 Activity

Protein fractions of aVSMC stimulated for 24 and 48 h were obtained after 30 min incubation on ice with 100 µL cell lysis buffer and used to determine caspase-3 activity (*n* = 5 − 6). A total of 60 µL of the sample were incubated with 60 µL of Caspase-reaction buffer and 42 µM DEVD-AFC (end concentration) for 90 min at 37 °C. The fluorescence of the cleaved product AFC was measured with a plate reader (Infinite M200, Tecan, Crailsheim, Germany) at 400 nm excitation and 505 nm emission wavelengths. Cleaved AFC was quantified using a calibration curve with known AFC concentrations and normalized to the total amount of protein contained in the sample (determined by BCA assay).

### 2.10. Senescence-Associated β-Galactosidase Measurement

Measurement of the SA-β-gal activity was performed on aVSMC that underwent up to 4 cell culture passages (*n* = 6). The protocol was adapted from Debacq-Chainiaux et al. [27]. Shortly, aVSMC were cultivated close to confluency in a 96-well plate and incubated with EGF and/or U46619. After 48 h incubation, the cells were stained with Hoechst 33,342 (final concentration: 5 µg/mL) and C12FDG (final concentration: 33 µM) in HEPES-Ringer buffer for 2 h at 37 °C, 5% CO_2_. After three washes with 1xPBS, the cells were fixed with 4% formaldehyde for 30 min at room temperature. Cells were washed again three times with 1xPBS. Hoechst 33,342 (Excitation/Emission: 358/461 nm) and C12FDG (490/514 nm) signals were detected by digital fluorescence microscopy (Cytation3, BioTek, Bad Friedrichshall, Germany). Data analysis was performed with Gen5 2.09 software (BioTek, Bad Friedrichshall, Germany). The values for each independent replicate correspond to the mean of 6 wells measured per treatment condition.

### 2.11. ELISA for 5-Bromo-2′-deoxyuidine

aVSMC were cultivated close to confluency in a 96-well plate and incubated with 10 µM (final concentration) BrdU and stimuli (EGF and/or U46619) (*n* = 6). Cells incubated with BrdU and media with serum and supplements served as the positive control. After 24 h incubation, the cells were fixed with 4% formaldehyde for 60 min at room temperature and then washed 3 × 5 min (washing step) with permeabilization buffer. Cell DNA was denatured with 2N HCl for 30 min at room temperature. After a new washing step, cells were incubated at room temperature for 2 h in a blocking solution. Cells were incubated overnight at 4 °C with anti-BrdU antibody diluted in 5% BSA/permeabilization buffer. The following day, after a washing step with 1xPBS, cells were incubated for 2 h at room temperature with Anti-mouse IgG, HRP-linked antibody in 5% BSA/permeabilization buffer. Followed by a washing step with permeabilization buffer and one with 1xPBS. Cells were then incubated with HRP-Substrate for 15 min at room temperature. The reaction was stopped with 1M H_2_SO_4_, and the absorption was measured immediately at 490 nm (corresponds to BrdU incorporation). To be able to normalize these data to the cell density of each well, the cells were washed with 1xPBS and stained with 0.2% Trypan Blue for 5 min at 37 °C. Washing steps with 1xPBS were repeated until the washing solution remained clear. Finally, the cells were incubated with 1% SDS for 30 min at room temperature with gentle shaking, and the absorption was measured at 560 nm. The values for each independent replicate correspond to the mean of 6 wells measured per treatment condition.

### 2.12. Osteopontin Secretion Measurement

Cell culture media was collected after 48 h incubation with EGF and/or U46619. Cells were lysed with RIPA buffer (as described in Western Blot section), and the protein concentration was determined with BCA assay. Ultra-15 centrifugal Filter Devices 10 K (Amicon-Merck, Darmstadt, Germany) and a “Mouse Osteopontin ELISA kit” were used to concentrate the media (concentration factor: 15–25) and determine the Osteopontin concentration, respectively, following the manufacturers’ instructions (*n* = 4).

### 2.13. Statistical Analysis

For all laboratory confirmation experiments, significant differences between groups were assessed by Wilcoxon test (*p* < 0.05) and Chi-squared test for outlier removal was performed with the outliers R package (version 0.14—https://cran.r-project.org/package=outliers (accessed on 19 May 2022)).

## 3. Results

### 3.1. aVSMC Express Genes Encoding for AT1R, EGFR and TP but Respond to Exclusive EGF and U46619-Stimulation Only

In mice, AT1R, EGFR and TP are encoded by the genes *At1ra* and *At1rb* (isoforms), *Egfr* and *Tbxa2r*, respectively. RNA-sequencing data and ddPCR experiments showed that aVSMC express *At1ra* (predominant AT1R isoform [4]) but at a low level in comparison to *Egfr* and *Tbxa2r* (Appendix A). *Tbxa2r* is the only gene out of the three found regulated at the mRNA level throughout the different conditions (*log2*-fold change = −1.49 and −2.02 after 24 h incubation with U46619 alone or combined with EGF, respectively—Appendix A). Additionally, EGFR protein expression level was decreased by EGF (Appendix A), and this effect was slightly refrained by the addition of U46619. The protein expression of AT1R and TP could not be measured due to a lack of antibodies with sufficient specificity for these targets.

According to the literature, direct EGFR activation or its transactivation trigger EGFR and ERK1/2-phosphorylation [28]. These parameters were therefore used to check the responsiveness of aVSMC to the different stimuli. EGFR-phosphorylation (Y1068) status increased after 10 min incubation with EGF but did not change with AngII or U46619 (Appendix A). However, the EGF-induced EGFR phosphorylation was significantly enhanced by the activation of TP with U46619. AngII also tended to increase it but without reaching the statistical significance threshold (*p* = 0.28). Additionally, all incubation types except AngII alone triggered an increase in ERK1/2-phosphorylation (Appendix A). These results suggest that aVSMC were responsive to EGFR and TP but not to exclusive AT1R-activation.

### 3.2. Exclusive EGFR and TP but Not AT1R-Activation Lead to Time-Dependent Gene Expression Regulation in aVSMC

Differential expression analyses were performed on data from RNA collected after 24 and 48 h incubation with EGF, AngII or U46619, in order to identify the effect of each receptor on aVSMC transcriptome. Figure 1a displays the number of differentially expressed genes (DEG) found for each condition. Incubation with AngII alone did not trigger gene expression regulation after neither 24 nor 48 h. However, incubation with EGF and especially with U46619 did, as highlighted by gene expression clustering heatmaps (Figure 1b,c and Appendix A). In more detail, EGF incubation led to the identification of 326 DEG after 24 h incubation but of none after 48 h, whereas 3796 and 570 DEG were found after 24 and 48 h incubation with U46619, respectively. Thus, exclusive EGFR and TP-activation affected aVSMC transcriptome but in a transient manner.

### 3.3. Simultaneous Receptor Activation Leads to Qualitative and Quantitative Synergistic Gene Expression Regulation

Differential expression analyses were also performed for combined incubations (EGF and/or AngII and/or U46619 vs. control). The combination of EGF with AngII or/and with U46619 led to overadditive amounts of DEG in comparison to individual incubations after 24 h (Figure 1a). AngII exerted no effect on its own but more than doubled the number of genes affected by EGF, indicating a permissive effect. Similarly, U46619 and EGF acted synergistically after 48 h since EGF enhanced the number of genes affected by TP activation without having an effect on its own.

We compared the lists of DEG identified for each type of incubation after 24 h and annotated intersects that comprised synergistically regulated genes (Figure 2a). For instance, some genes required the combination of at least two stimuli to be regulated (qualitative synergism—stimulation), while the effect of a substance on some genes was blocked by the addition of another one (qualitative synergism—inhibition). In addition, Figure 2b shows the *log2*-fold change distributions of the DEG comprised in all compared lists (intersection i7). A significant shift in the distribution was induced by the combination of EGF and U46619 but not by the addition of AngII. The same observations were made for all intersections with more than 50 genes (Appendix A). These results indicate a global quantitative synergistic effect of EGF and U46619 but not of “EGF and AngII” or “AngII and U46619” after 24 h. Thus, this suggests that the combined EGFR and TP-activation affect the amplitude of the gene expression changes. The comparison of measured and expected fold changes (after combined incubation) for the genes comprised in each intersection confirmed this observation (see Methods and Appendix A). Intersections that included genes for which the combination of stimuli led to an overadditive or an underadditive effect (quantitative synergism) were also highlighted in Figure 2a.

The analysis of the 48 h datasets only identified DEG regulated by U46619 alone or combined with EGF. A total of 414 genes required the activation of both receptors to be regulated (qualitative synergism), and 79 were regulated only by U46619 alone (EGF-inhibiting effect) (Figure 3a). The *log2*-fold change distributions for the genes comprised in the intersection (491 genes) did not show any significant shift (Figure 3b). This means that U46619 alone is sufficient to trigger the maximum response for these genes and that there is a qualitative but no global quantitative synergistic effect of EGF and U46619 after 48 h anymore, supporting the previous observation of a time-dependent response.

### 3.4. The Putative Regulators of the EGF and U46619-Induced Changes in Gene Expression Play a Role in Cell Proliferation and Cell Death

Upstream analysis with IPA allows predicting potential regulators (e.g., transcription factors, cytokines, enzymes) of the observed changes in gene expression (based on information from the curated literature). Here, only the lists of genes regulated by EGF, U46619 or both substances after 24 h were used as inputs since these conditions explained most of the gene expression regulation (Figure 1a). The obtained list of significantly enriched regulators (Appendix A) was screened and divided into three categories: (1) EGF, (2) U46619 and (3) EGF and U46619-specific (Table 1 and Appendix A). The latter include IKBKG, KDM5B, MAX, PCLAF, PTGER2, S100A6 and TFEB, which have been described as involved in cell proliferation [29,30,31,32], cell death [33,34] and senescence [30,35] regulation.

### 3.5. Genes Regulated by EGF and/or U46619 Are Enriched in Pathways Related to Cell Cycle, Cell Death and Metabolism

Because EGF and U46619 have time-dependent individual and synergistic effects on aVSMC transcriptome, we also compared the lists of DEG obtained after 24 and 48 h. First, all lists of DEG but the one corresponding to the combination of the three substances (not available for 48 h) were matched (Figure 4a). This resulted in the division of the DEG into three categories: (1) genes regulated after 24 h only (“early” effect), (2) genes regulated after 48 h only (“late” effect) and (3) genes regulated throughout the activation timeline (“lasting” effect).

U46619 alone or combined with EGF were the only two conditions that triggered significant gene expression regulation after 48 h. A second comparison was, therefore, made, only considering the DEG regulated by these two types of incubation (Figure 4b), highlighting again a response in three phases (early, late or lasting). Because the analyses for the 24 and 48 h datasets were run in parallel due to the batch effect (Appendix A), only the sign but not the values of the *log2*-fold change can be compared for the genes comprised in the intersections. Doing so, most concerned genes appeared to be regulated in the same direction after 24 and 48 h (Appendix A).

GO term enrichment analysis was used to identify pathways and cellular mechanisms potentially influenced by the EGF, U46619 and AngII-induced gene expression regulation. The list of genes included in each intersection of the UpSet plot (Figure 4a) and in the different parts of the Venn diagrams (Figure 4b) served as input. In order to apply an appropriate threshold to identify significantly enriched terms, GO term enrichment analysis was simulated with sets of random genes (picked in the whole mouse Ensembl v101 annotation) of the same size as the UpSet plot intersections. The simulation was run 100 times and the minimally adjusted *p*-values obtained each time were extracted (Appendix A). All minimal *p*-values were below 0.05, independently of the set size, meaning that this commonly used threshold was not stringent enough here. On the contrary, most of the simulations with random genes did not yield *p*-values below 0.001. Therefore, our actual data were filtered using this threshold in addition to an enrichment-based one (with *E* ≥ 3—see Methods). The enriched terms obtained after filtering and removal of the redundant ones are summarized in Figure 4c (detailed results in Appendix A). These results confirmed a time-dependent response to the stimuli: early regulated genes were enriched in terms mostly related to cell cycle and DNA replication, while the late regulated ones were enriched in cell death-related pathways. Meanwhile, the continuously regulated genes showed enrichment in metabolic processes (e.g., glucose, ATP, pyruvate metabolism) or in senescence-related pathways.

### 3.6. Simultaneous EGFR and TP-Activation Regulate Their Respective Effect on Cell Cycle and Cell Senescence

As seen in Figure 4c, genes regulated after 24 h incubation were enriched in GO terms mostly related to cell cycle, DNA replication and maintenance. On the contrary, the few enriched pathways with genes regulated after 48 h hinted at cell death regulation. These results incited the investigation of the roles of each receptor in aVSMC cell proliferation and survival, especially as these processes have been associated with vascular injuries and diseases [36,37,38]. These measurements focused on the effect of EGF and U46619 but not AngII, as we previously observed that aVSMC did not respond to it.

Caspase-3 served as an indicator for cell death and more, especially apoptosis. However, incubation with EGF or/and U46619 for 24 or 48 h did not lead to any significant change in its activity (Figure 5a).

DNA synthesis and thus cell cycle were estimated by BrdU-incorporation measurement (Figure 5b). Individual incubation with EGF and U46619 led to an increase and a decrease in DNA synthesis, respectively. The combination of both substances did not have any significant effect anymore, in part due to a high variation between the independent replicates. This suggests that EGF and U46619 may cancel each other’s effect.

Finally, senescence-associated β-gal (SA-β-gal) acted as a measure for cell senescence (Figure 5c–e). The measurements were made on aVSMC that underwent less than five passages to avoid replicative senescence (caused by long cell culture periods of primary cells). The proportion of senescent cells (objects defined as green) increased with U46619, but no significant change was triggered by EGF alone or combined with U46619. This observation implies that EGF may also modulate the effect triggered by U46619 concerning cell senescence.

### 3.7. EGFR and TP-Activation Synergistically Regulate Genes Relevant in Cardiovascular Injuries and Diseases

Previous analysis steps gave us the opportunity to identify which cellular processes may be influenced by EGF and/or U46619. In addition to this global approach, we also adopted a targeted one to identify putative key actors in the observed phenotypical changes. We thus filtered genes for which the combination of two stimuli triggered a stronger effect than just one. To do so, a pairwise comparison (*t*-test) of normalized counts (FPM) was made for all conditions at both time points. Although this approach may have resulted in underestimating the number of recognized genes due to reduced statistical strength, it had the advantages of having a constant reference group and of working with unvarying values, which would not have been the case with differential expression analyses comparing the different single and combined incubations.

The genes comprised of the intersections of the UpSet plot (Figure 4a) were thus screened with this method. We selected genes that showed a significant difference (*p* < 0.05) when comparing the single incubations to the corresponding double one (e.g., comparing EGF vs. EGF and U46619 and comparing U46619 vs. EGF and U46619) for at least one time point. Genes comprised of intersections associated with clear qualitative synergism or that did not comprise DEG regulated by the concerned conditions were filtered out. 

The genes that passed these filters are summarized in Table 2 (individual values shown in Appendix A). EGF and U46619 triggered stronger regulation but with low amplitude for most of the involved genes. However, some of these showed a clear synergistic effect. *Klf15* (Kruppel-like factor 15) and *Spp1* (Secreted Phosphoprotein 1, codes for Osteopontin—OPN) were among the latter (Figure 6a). The products of these two genes have been associated with vascular injuries and diseases [39,40,41].

*Klf15* expression was regulated during the early response (intersection i7 in Figure 4a). Indeed, differentially expression analyses showed *Klf15* to be significantly down-regulated by EGF or/and U46619 after 24 h only (Appendix A), with a stronger response induced by the combined stimuli (*log2*-fold change = −2.26 when compared to control). qPCR on independent samples confirmed the regulation after 24 h (Figure 6b). Nevertheless, it also showed that this effect of EGF and U46619 lasted after 48 h, but without a visible synergistic effect of the two substances anymore. The lack of specific antibodies prevented the measurement of *KLF15* protein expression.

On the contrary, *Spp1* was continuously (lasting effect, intersection i17 in Figure 4a) and strongly up-regulated by the combination of EGF and U46619 in the RNA-sequencing data (*log2*-fold change = 2.89 and 3.61 after 24 and 48 h, respectively). qPCR confirmed the severe up-regulation by EGF and U46619 (10-fold increase in comparison to the control group) at the mRNA level (Figure 6b). Additionally, ELISA measurement confirmed that EGF and U46619 also triggered a strong up-regulation (7-fold increase) of the secreted OPN (Figure 6c).

## 4. Discussion

In this study, we aimed to show that relevant assessments have to include the investigation of a receptor as a function of the activation status (on/off) of other receptors, as having models close to physiological or pathophysiological situations allows more reliable observations and conclusions to be made. We wanted a health-wise pertinent and controlled in vitro model to do that. We, therefore, focused on aVSMC and their inherent functional interaction of EGFR, AT1R and TP, which are relevant receptors in the cardiovascular system physiology and related diseases. RNA-sequencing was used as an unbiased and global readout to assess the long-term effect of each receptor and of their crosstalk on gene expression.

Evidence of functional interaction of EGFR with TP and especially with AT1R concerning the regulation of phenotypical changes in the cardiovascular system (e.g., blood pressure regulation, wall stiffening) already exists [13,14,15]. Additionally, some studies hint at a possible synergistic response subsequent to the activation of EGFR and of one of the other two receptors, including for cell proliferation [42] or DNA synthesis [43,44] regulation. However, the underlying mechanisms and particularly the changes in gene expression that result in these phenotypical changes remain unclear. To our knowledge, this study is the first one that fills this gap by providing transcriptomic data from primary vascular smooth muscle cells following the activation of the various receptors of interest.

Moreover, by focusing on moderate and late time points of their activation timeline, our data capture the long-term effects of receptor activation on gene expression, which is relevant to explaining phenotypical changes and pathophysiological mechanisms. Incidentally, aVSMC showed a time-dependent response to receptor activation, with less DEG identified after 48 h than 24 h and different predicted cellular consequences by the functional analysis at the two time points. A change in aVSMC ability to respond to the different stimuli (given in saturating amount) may have been at the origin of these variations as close to no EGFR remained after 24 h incubation with EGF, and the mRNA expression of TP was halved after 24 h incubation with U46619. This suggests that significantly less EGFR and TP are available to convey the signal into the cells already after 24 h of activation.

While we observed gene expression regulation upon EGFR or TP-activation, aVSMC did not show signs of responsiveness to exclusive AT1R-activation. Because AT1R (encoded by *Atr1a* and *At1rb* in rodents) was barely detectable in our RNA-sequencing data, a finding confirmed for *At1ra* (main source of AT1R [4]) by ddPCR, we suppose that the lack of responsiveness to the stimulation with AngII alone was linked to the low expression level of its receptor. Nevertheless, the detection of mRNA coding for AT1R, although at a low level, enabled us to pursue our study. Indeed, for lack of effects on gene expression of AT1R activation per se, it was of interest to determine whether AT1R activation affects gene expression in combination with other receptors. AngII turned out to have a positive modulatory effect in the presence of activated EGFR, leading to a trend towards increased EGFR phosphorylation (but not ERK1/2 phosphorylation) and to a higher number of genes regulated by EGF (qualitative synergism) after 24 h. Thus, AngII does not have an effect on its own but intensifies the effect of EGF, likely via one of its ERK1/2-independent associated pathways [4]. On the contrary, there was no detectable interaction of AT1R with TP.

Meanwhile, the combination of EGF and U46619 led to synergistic gene expression regulation in both qualitative and quantitative manners. This is particularly striking after 48 h, where the synergistic response elongates the effect of EGF as it has no effect on its own anymore but still potentiates the effect of U46619. This lasting overadditive response to the simultaneous EGFR and TP-activation may partially result from the slightly but significantly reduced EGF-induced degradation of EGFR in the presence of U46619 (suggesting that more EGFR remains available at the cell surface for further signal transduction) and from the enhanced EGFR phosphorylation. In addition, bioinformatics analysis predicted that some putative key regulators of the gene expression regulation are specifically activated by the simultaneous receptor activation. Taken together, these results imply that the synergistic gene expression regulation that follows the simultaneous activation of EGFR and TP may result both from the amplification of existing downstream cascades and from the activation of distinct ones.

The results of the functional analysis also support this observation. Indeed, while the genes regulated only upon the simultaneous activation of two receptors (qualitative synergism) were not enriched in pathways related to distinct cellular processes in comparison to the genes indifferently regulated by single or double incubations, they were also significantly enriched in pathways related to similar processes, suggesting a putative additional or distinct effect specifically induced by the double incubation. For instance, genes regulated upon the individual or combined activation of EGFR and TP, but also genes regulated specifically by their combined activation, were significantly enriched in pathways related to cell cycle and DNA replication. We could experimentally confirm the alteration of DNA synthesis by EGF and U46619 alone, which led to an increase and to a decrease in BrdU incorporation in aVSMC, respectively. However, the incubation with both EGF and U46619 had no effect (no significant change in BrdU incorporation). Previous studies showed that EGFR-activation is associated with VSMC proliferation [15,45], and it is generally accepted that TP-activation promotes it as well. Nevertheless, the published results on this point show contradictions. For instance, U46619 was found to increase DNA synthesis in human bronchial VSMC cultivated with 1% serum [46] and in rat aVSMC cultivated in serum-free media, which was enhanced by the addition of platelet-derived grown factor (PDGF) [47]. However, another study reported that U46619 alone was not sufficient and that PDGF was actually required to trigger DNA synthesis in serum-free bovine coronary VSMC [48]. On the other hand, one study showed that U46619 triggered the cell cycle exit of serum-free piglet pulmonary artery monocytes [49]. Therefore, the role of TP in cell proliferation regulation appears highly context-dependent, and the published results suggest an ability to potentiate the effect of growth factors rather than a mitogenic effect. In that respect, concerning the regulation of DNA synthesis by the combined TP and EGFR-activation in particular, Kong et al. [43] previously reported that U46619 potentiates the EGF-induced ^3^H-thymidine incorporation in smooth airway muscle cells. However, we could not find any information regarding VSMC and our results suggest that the individual effects of EGF and U46619 on DNA synthesis cancel each other in aVSMC. This may result from a compensatory effect of the genes regulated by each substance or from the synergistic gene expression regulation (with the regulation of a distinct group of genes).

Additionally, regulated genes by U46619 alone or in combination with EGF were significantly associated with cellular senescence. The latter is an aging process during which cells stop proliferating but do not die [50]. Senescent cells also undergo various phenotypical changes, such as an increase in the endogenous β-gal activity, then called SA-β-gal. We observed that the incubation with U46619 did not change aVSMC apoptosis rate but led to a reduction in DNA synthesis and an increase in SA-β-gal. This suggests that TP-activation is associated with senescence development in aVSMC. Nevertheless, while EGF does not appear to regulate cell senescence on its own, its addition led to an inhibition of the U46619-induced effect.

Therefore, regarding DNA synthesis and cellular senescence, the downstream effects of EGFR and TP-activation appear to modulate rather than potentiate each other. Because a shift from a contractile phenotype to a proliferative one and cellular senescence has been both associated with vascular diseases [51,52], this equilibrium between EGFR and TP-activation consequences may, therefore, lead to a protective state in aVSMC.

Some of the predicted key regulators of gene expression following the simultaneous activation of EGFR and TP have also been described as regulators of DNA synthesis, cell cycle and cell senescence. For instance, the histone demethylase KDM5B promotes pulmonary arterial VSMC proliferation [29] and fibroblasts senescence [35]. On the other hand, S100A6 was described as a repressor of senescence traits [30]. The activity of these proteins is predicted to be down and up-regulated, respectively, supporting the thesis of a balanced effect of the two substances. Further experiments to evaluate the consequences of EGFR and TP-activation on the activity of these regulators are still required.

Finally, our targeted approach highlighted two vascular disease-related genes synergistically regulated by EGF and U46619. *Klf15* encodes for a transcription factor of the same name. It has been identified as an inhibitor of cell proliferation in VSMC [39] that is strongly expressed under basal conditions but significantly reduced after injury [40]. It was down-regulated by EGF and/or U46619, suggesting an increase in cell proliferation for all conditions. We thus could not link the phenotypical changes we observed and KLF15. Nevertheless, KLF15 can have pleiotropic effects as a transcription factor, and further studies are needed to identify its exact role in this context.

OPN (*Spp1*) was reported to be regulated downstream of EGFR phosphorylation [53]. Its overadditive regulation by EGF and U46619 could, therefore, result from the double stimuli-dependent enhanced EGFR phosphorylation we observed. The role of OPN in vascular diseases is highly controversial and has been profusely discussed over the last years [41,54,55]. For instance, it is used as a biomarker of vascular remodeling due to its high concentration in plasma from patients with cardiovascular diseases and in vessels from animal models for atherosclerosis. However, many in vivo studies demonstrated its ability to inhibit vascular calcification, suggesting that the high OPN concentrations are actually a protective mechanism against advanced stages of vascular diseases such as atherosclerosis. Additionally, evidence suggests that OPN may have different functions depending on post-transcriptional modifications, its concentration and through time. While we could not directly relate the synergistic up-regulation of OPN to the phenotypical changes we monitored, we propose that OPN may be involved in the putatively protective EGFR-TP system.

## 5. Conclusions

In conclusion, the simultaneous activations of EGFR and AT1R or TP enhance the response to the former concerning gene expression regulation in aVSMC, including for genes involved in the cardiovascular system maintenance. However, the phenotypical changes downstream of EGFR and TP-activation appear to balance, implying physiological feedback in pathways downstream of the two receptors. Thus, although additional experiments are still needed to confirm the hypotheses generated here, and even though both EGFR and TP-activation are usually associated with harmful outputs in the cardiovascular system, this first evidence suggests a protective EGFR-TP system in aVSMC. This study and its results, therefore, highlight the importance of using unbiased approaches that are as close to physiology as possible in order to draw accurate conclusions concerning the role of a given receptor.

## Figures and Tables

**Figure 1 cells-11-01936-f001:**
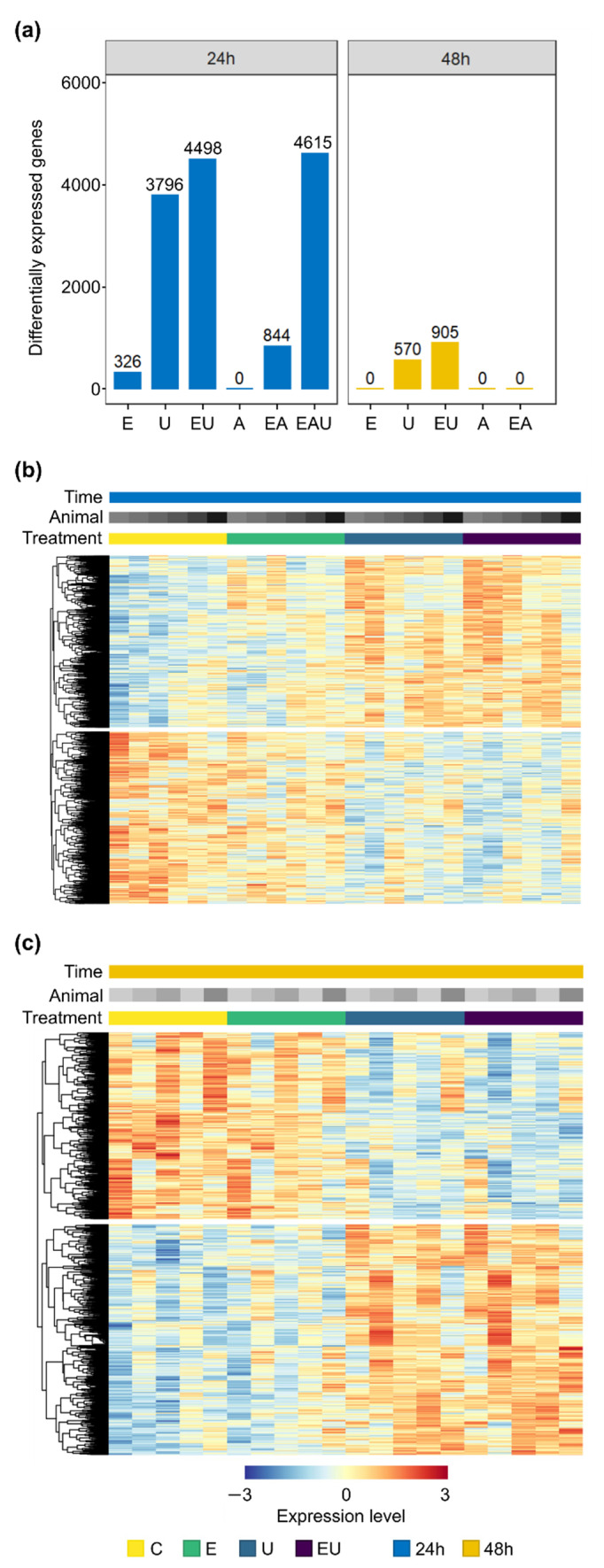
**EGF and U46619 lead to gene expression regulation in aVSMC but not AngII.** E: EGF, U: U46619, EU: EGF and U46619, A: AngII, EA: EGF and AngII, EAU: EGF and AngII and U46619. (**a**) Number of differentially expressed genes (DEG) after 24 and 48 h incubation (compared to the respective control groups). (**b**,**c**) Heatmaps showing the normalized expression (*log* scale) of genes identified as significantly regulated for at least one comparison (EGF or U46619 or EGF and U46619-treated vs. control group) in the analyses for 24 h (**b**) or 48 h (**c**) incubation. A total of 4921 and 984 genes were thus included for each time point. Each row represents a gene, and each column a sample. Expression levels were additionally row-wise centered (subtraction of the mean from each value) and scaled (division by the standard deviation). Rows were clustered based on Euclidean distance (complete method, calculated by pheatmap, version 1.0.12—https://cran.r-project.org/package=pheatmap (accessed on 19 May 2022)). Heatmaps comprising all conditions are available in Appendix A.

**Figure 2 cells-11-01936-f002:**
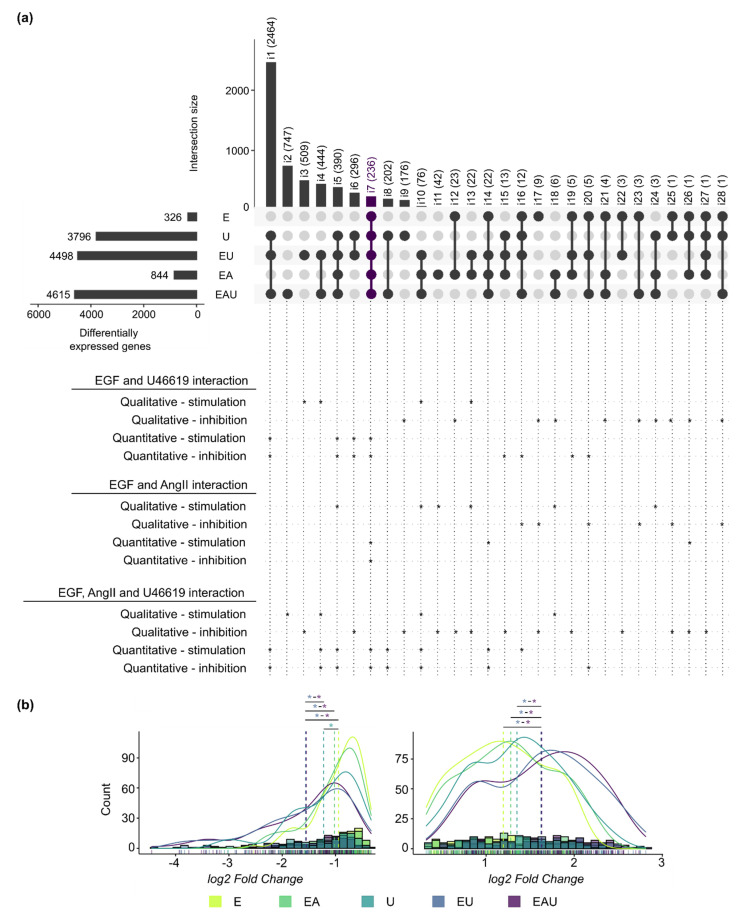
**EGFR and TP act, in part, synergistically on gene expression regulation after 24 h (next page).** (**a**) The overlaps of the lists of DEG after 24 h incubation are presented in an UpSet plot. Each row corresponds to a set of DEG and each column to one segment of a hypothetical Venn diagram. A black or a grey dot indicates that this set is included or not in this intersection, respectively. The names (i1 to i28) and the number of comprised genes are indicated for each intersection. The interactions between EGF, AngII and U46619 are indicated below the UpSet plot. Asterisks (*) mark intersections comprising genes regulated in a synergistic manner. “Qualitative-stimulation” and “-inhibition” means that an additional stimulus leads to or blocks the regulation of a new set of genes, respectively, in comparison to a simpler incubation. On the other hand, “quantitative-stimulation” or “-inhibition” means that genes had significantly different (*p* < 0.05) expected and measured fold changes when comparing single and multiple incubations (see Methods and Appendix A). (**b**) Histograms and density curves for the *log2*-fold changes (calculated by edgeR) of the genes comprised in intersection i7 (left: down-regulated genes, right: up-regulated genes). The dotted lines correspond to the mean. The Kolmogorov–Smirnov test was used to assess if the shifts in distribution were statistically significant (* *p* < 0.05. The color of the asterisks corresponds to the treatment with which the comparison was made.). Detailed *p*-values and histograms for the other intersections are shown in Appendix A.

**Figure 3 cells-11-01936-f003:**
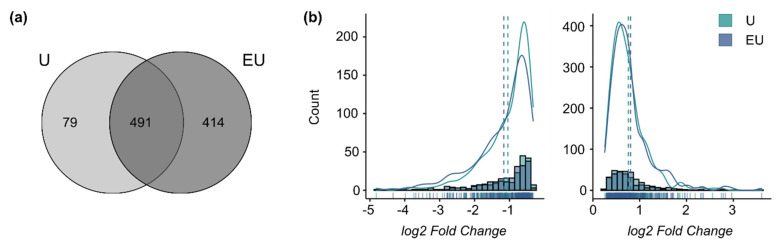
**EGFR and TP-activation triggers a qualitative but no quantitative synergistic regulation after 48 h.** (**a**) Overlap of the lists of DEG after 48 h incubation. (**b**) Histograms and density curves for the *log2*-fold changes (calculated by edgeR) of the genes comprised in the intersection of the Venn Diagram. (left graph: down-regulated genes, right graph: up-regulated genes). The dotted lines correspond to the mean. The Kolmogorov–Smirnov test was used to assess if the shifts in distribution were statistically significant.

**Figure 4 cells-11-01936-f004:**
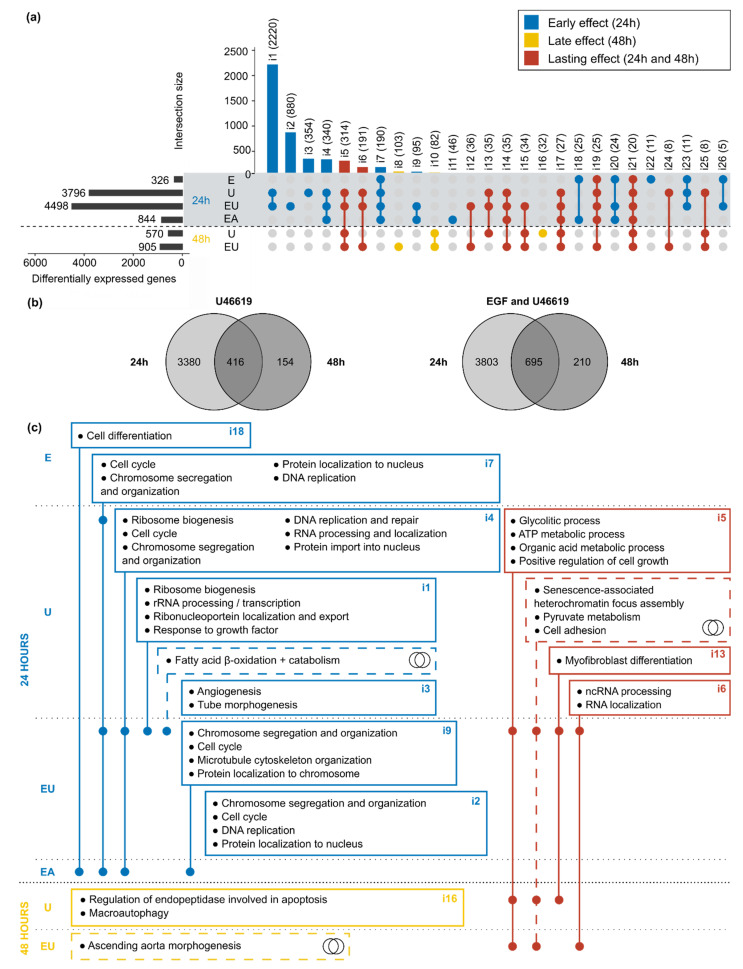
**EGFR and TP putative regulation of cell cycle, cell death and metabolism-related pathways are time-dependent.** (**a**) Overlaps of all lists of DEG after 24 and 48 h incubation. Only intersections comprising at least 5 genes are displayed. The name (i1 to i26) and the number of comprised genes are indicated for each intersection. The intersections are divided into three subgroups corresponding to “early” (DEG after 24 h incubation), “late” (48 h) and “lasting-effects” (24 and 48 h). (**b**) Direct comparison of the lists of DEG after 24 and 48 h for the two conditions (U46619 and “EGF and U46619”) that led to gene expression regulation at both time points. (**c**) The genes comprised in each intersection of the UpSet plot (Figure 4a) and in the different parts of the Venn diagram (Figure 4b) were used as inputs for GO term enrichment analysis. The figure displays the main enriched terms and is based on the same principle as an UpSet plot, with a dot in each concerned row. The intersection numbers of the UpSet plot or miniature Venn diagram symbols indicate which lists of genes were used as the input.

**Figure 5 cells-11-01936-f005:**
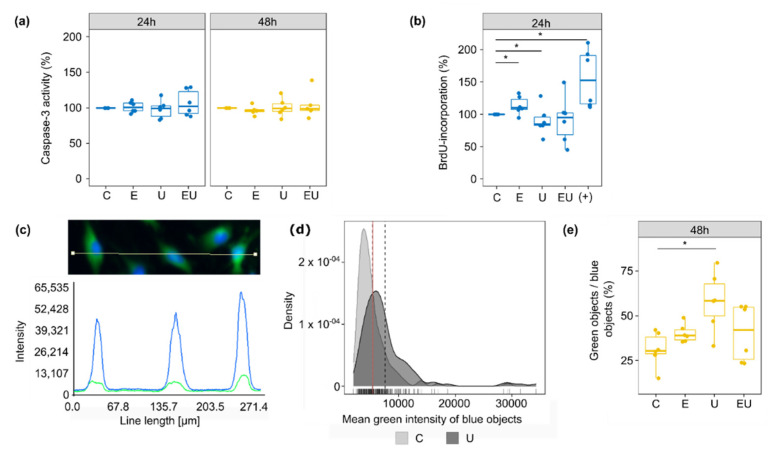
**Effect of EGFR and TP receptor activation on apoptosis, cell cycle and senescence.** (**a**) Caspase-3 activity was measured after 24 and 48 h incubation with EGF and/or U46619. The control condition (media without stimuli) was used as a reference for each independent replicate (*n* = 5–6). (**b**) DNA synthesis rate was measured with BrdU incorporation after 24 h incubation with EGF and/or U46619. The supplemented media used for the usual cultivation of aVSMC (+) served as a positive control. The control condition (media without stimuli) was used as a reference for each independent replicate (*n* = 6). (**c**–**e**) β-gal activity was measured in the green channel (after staining with C12FDG) of a fluorescence microscope, while Hoechst staining (blue channel) permitted the identification of cell nuclei (blue object) (*n* = 6). (**c**) Because the green signal mostly co-localized with the blue one, the β-gal activity of a single cell was defined as the mean green intensity of the corresponding blue object. (**d**) In order to determine which objects showed an enhanced β-gal activity (senescent cells), the mean green intensity of blue objects was plotted for the control (lower signal) and U46619-treated (brighter signal) groups (density plots). The grey dashed lines and red solid one depict the mean for each distribution and the intersection of the two distributions, respectively. The value at the crossing of the red line on the *x*-axis was used as the cut-off to determine if an object was greener than randomly expected (green objects). This threshold was determined for each independent experiment. Each time, the Kolmogorov–Smirnov test was used to ensure that the shifts in distribution were significant (*p* < 0.05). (**e**) The fraction of senescent cells was defined as the percentage of blue objects that were also green. * Wilcoxon test *p* < 0.05.

**Figure 6 cells-11-01936-f006:**
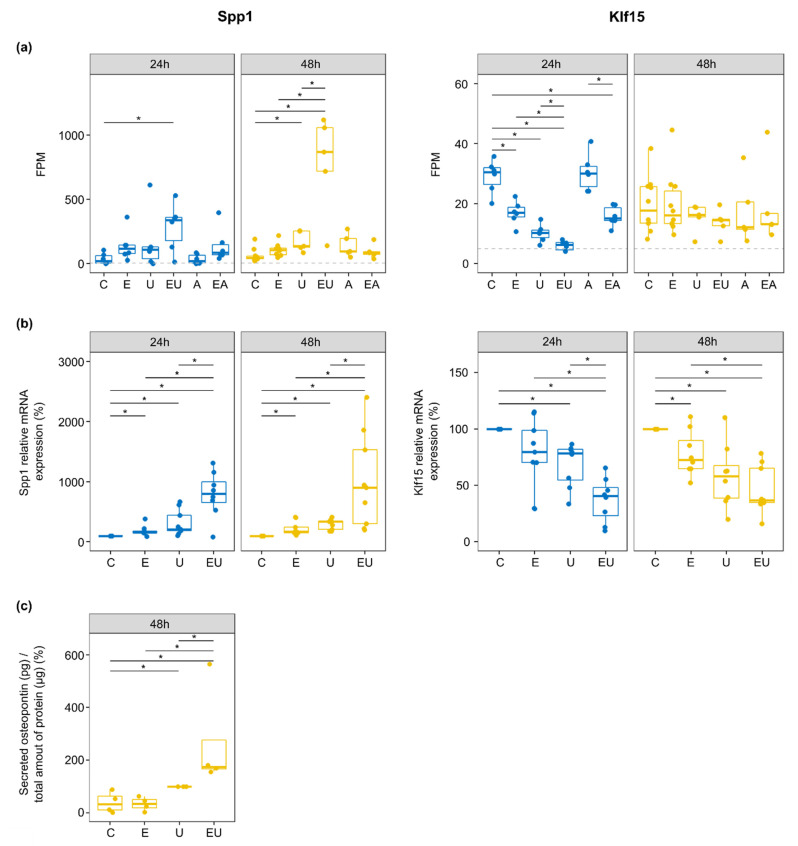
**EGFR and U46619 synergistically regulate two cardiovascular relevant genes, *Spp1* and *Klf15*.** (**a**) FPM for *Spp1* and *Klf15* genes. The dotted line corresponds to the 5 FPM threshold employed to filter out lowly expressed genes (* *p* < 0.05—*t*-test). (**b**) Relative mRNA expression of *Spp1* and *Klf15* measured by qPCR. Outliers were removed, and the control group was used as a reference for each independent replicate. (*n* = 8–9, * *p* < 0.05—Wilcoxon test). (**c**) The amount of secreted OPN after 48 h incubation was measured by ELISA and normalized to the total amount of protein isolated from the cells cultivated with the measured media. The U46619-treated group served as a reference as the values for the control groups were close to the background ones (*n* = 4, * *p* < 0.05—Wilcoxon test).

**Table 1 cells-11-01936-t001:** **Predicted regulators of the changes in gene expression induced by EGF or/and U46619.** IPA software-calculated Z-scores based on annotations in its internal database. Negative and positive Z-scores correspond to predicted enhanced and inhibited the activity of the regulator, respectively. N/A stands for undetermined Z-scores either because there was no enrichment for this regulator for a given dataset (−log(adj-*p*) = 0) or because of a lack of information concerning the direction in which this regulator may influence enriched genes (-log(adj-*p*) ≠ 0). Predicted upstream regulators were filtered for EGF, U46619 and “EGF and U46619”-specific ones. To do so, the whole list of putative regulators (Appendix A) was filtered for Z-scores and adjusted *p*-values. For instance, to identify actors that may regulate gene expression only when incubated with EGF alone, the list was filtered on one hand for |Z-score_EGF_| ≥ 2 and –log(adj-*p*_EGF_) ≥ 3 (corresponds to adjusted *p*-value ≤ 0.001) and on the other hand for |Z-score_U46619_| < 2, |Z-score_EGF+U46619|_ < 2, –log(adj-*p*_U46619_) < 3 and –log(adj-*p*_EGF+U46619_) < 3. The same strategy was applied for the other categories using appropriate filters.

		EGF		U46619		EGF and U46619	
		Z-Score	−log(adj-*p*)	Z-Score	−log(adj-*p*)	Z-Score	−log(adj-*p*)
EGF-specific	IL2	2.12	3.20	1.32	1.32	0.92	1.14
	NLRP3	−3.00	3.64	−0.93	1.18	−1.73	0.97
U46619-specific	EIF2AK3	N/A	0.00	3.08	4.21	1.86	2.18
	PNPT1	N/A	0.00	3.34	3.80	N/A	0.00
EGF and U46619-specific	IKBKG	N/A	0.00	−1.79	2.45	−2.18	3.92
	KDM5B	−1.97	1.78	−1.75	0.83	−2.64	4.13
	MAX	N/A	1.21	N/A	1.82	2.00	4.04
	PCLAF	N/A	0.00	N/A	0.00	3.16	3.37
	PTGER2	N/A	0.00	N/A	0.00	4.70	6.61
	S100A6	1.07	2.07	1.71	0.74	2.35	3.02
	TFEB	N/A	0.00	−1.92	2.30	−3.92	3.81

**Table 2 cells-11-01936-t002:** **Quantitative synergistic effect on gene expression.** FPM comparison highlighted genes for which the incubation with combined substances led to a significant increase in the effect magnitude. Genes with *p* < 0.05 (*t*-test) when comparing EGF to “EGF and U46619” and U46619 to “EGF and U46619” were selected. “Substances” and “regulation” columns indicate which combination triggered a synergistic effect and in which direction.

Gene	Ensemble Identifier	Intersection	Substances	Regulation
*Ctdsp1*	ENSMUSG00000026176	i1	EGF, U46619	Down
*Fam110b*	ENSMUSG00000049119	i5	EGF, U46619	Down
*Gm6665*	ENSMUSG00000091561	i17	EGF, U46619	Down
*Gstm2*	ENSMUSG00000040562	i17	EGF, U46619	Down
*Hadhb-ps*	ENSMUSG00000063684	i1	EGF, U46619	Down
*Inpp4a*	ENSMUSG00000026113	i1	EGF, U46619	Down
*Klf15*	ENSMUSG00000030087	i7	EGF, U46619	Down
*Mtap*	ENSMUSG00000062937	i1	EGF, U46619	Up
*Pcif1*	ENSMUSG00000039849	i1	EGF, U46619	Down
*Spp1*	ENSMUSG00000029304	i17	EGF, U46619	Up
*Srr*	ENSMUSG00000001323	i6	EGF, U46619	Down
*Tpst1*	ENSMUSG00000034118	i1	EGF, U46619	Down

## Data Availability

The analyzed data supporting the conclusions of this article are included within this article and its additional files. Additionally, raw RNA sequencing data are publicly available on Gene Expression Omnibus (GEO) database (https://www.ncbi.nlm.nih.gov/geo (accessed on 19 May 2022)). GEO accession number: GSE179389.

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
