# Peer review of "The Functional Interaction of EGFR with AT1R or TP in Primary Vascular Smooth Muscle Cells Triggers a Synergistic Regulation of Gene Expression"

_cells, 2022, doi:10.3390/cells11121936_

Round 1
Reviewer 1 Report
This manuscript try to investigate the impact on gene expression by activation EGFR and its crosstalk with AT1R or TP. It is an important question to investigate the alteration by one or multiple genes on qulitative or quantitative level.
Introduction Line 84 "We observed that exclusive EGFR- and TP- but not AT1R-activation led to gene expression regulation. " what this means? AT1R does not lead to gene expression regulation/alteration?
Line 150, what the FPM means? usually we used FPKM as the expression quantification, which will normalized by the gene length.
The replicates of experiments for RNA-seq were not described clearly in methods.
The "expected FC" in method is unclear. Why need this? Why this is so complex. For three replicates, If two FC >0 and one <0, that is not consistent and we should remove it. Line 164 the "-1" should be normal font size not subscript.
Line 273 coding->encoding, what is the U46619 stimulation? It should be described.
L277 how about At1rb?
L324 " Incubation with AngII alone did not trigger gene expression regulation after neither 24h nor 48h." It seems not reasonable.
fig1a for 48h, why no the EAU group?
What are the difference of the two figures in lower panel of fig2? It should be marked as a b c d for fig 2.
L397 Table 1, what the "regulators" mean? are they transcription factors? If not, how about the transcription factors changed in the stimulations? You may checked the AnimalTFDB database for TF reference.
The figure legend for fig 3 is unclear, especially for the fig 3b. what are the two figures? what are the lines with difference color? etc.
So for the results and conclusion of this paper, which kind of genes are qualitative synergy or quantitative synergy. These should be mentioned in the results and abstract.
Author Response
This manuscript try to investigate the impact on gene expression by activation EGFR and its crosstalk with AT1R or TP. It is an important question to investigate the alteration by one or multiple genes on qulitative or quantitative level.
Introduction Line 84 "We observed that exclusive EGFR- and TP- but not AT1R-activation led to gene expression regulation. " what this means? AT1R does not lead to gene expression regulation/alteration?
The activation of AT1R does not lead to gene expression regulation in our model. (This sentence was modified in the manuscript for more clarity – lines 84-85). We made this same observation after 24h and 48h, on independent replicates and in the two corresponding analysis runs made in parallel, showing the consistency of these results. We worked here with primary cells, what may explain the non-responsiveness to AngII, and with low AngII concentration (10nM, physiological concentration). We therefore concluded that the effect on gene expression by the activation of AT1R activation is limited and that it rather acts thought crosstalks with other pathways (e.g. EGFR) (discussed from line 586).
Line 150, what the FPM means? usually we used FPKM as the expression quantification, which will normalized by the gene length.
FPM stands for “Fragments Per Million” (added in the main text). Since we mostly use the FPM to makes comparisons between the conditions (control vs. treatments) but not to make comparisons between the genes themselves, the normalization by the gene length would have no real effect on the output.
The replicates of experiments for RNA-seq were not described clearly in methods.
The RNA samples used for sequencing were prepared in three different batches (due to the slow cell growth of primary cells in culture). We thus had 6 replicates (different animals) for the samples incubated for 24h (with EGF, AngII, U46619, EGF+AngII, EGF+U46619, or EGF+AngII+U46619), 5 replicates for the samples incubated for 48h (with EGF, AngII or EGF+AngII) and 5 replicates for the samples incubated for 48h (with EGF, U46619 or EGF+U46619). This issue has been clarified in the manuscript by rewording of lines 127-128.
The "expected FC" in method is unclear. Why need this? Why this is so complex. For three replicates, If two FC >0 and one <0, that is not consistent and we should remove it. Line 164 the "-1" should be normal font size not subscript.
This explanation is required as we want to explain how we calculated the expected FC (additive) and compare it to the measured FC (is it more than additive? Or less than additive? If so, we concluded of a synergistic effect of the concerned substances). The explanation was nonetheless shorten and slightly modified as we rather focused on the simultaneous activation of two receptors here (line 165).
Line 273 coding->encoding, what is the U46619 stimulation? It should be described.
U46619 is an agonist of the TP receptor that is commonly used for the activation of the latter in cell culture settings. This information is provided in the introduction (line 75).
L277 how about At1rb?
At1rb was very lowly expressed (only a few counts in the output of featureCounts and automatically removed by both DESeq2 and edgeR prior to the differential expression analyses). We therefore decided to focus on At1ra for the confirmation by ddPCR, as it had been described as the main isoform for AT1R and since it was still the only isoform expressed at a detectable level in our data despite its general low level of expression.
L324 " Incubation with AngII alone did not trigger gene expression regulation after neither 24h nor 48h." It seems not reasonable.
The incubation of our cells with AngII did not lead to any significant gene expression regulation, what could be explained by the already documented general lack of responsiveness of primary culture to AngII. Moreover, our results are consistent: the sample preparation (from different animals) and the data analyses were made separately for the 24h and 48h time-points and yet, the same lack of effect is observed. Finally, it is assumed that AngII leads directly to gene expression regulation but actually, only few data using the same concentration of AngII as us (10nM, physiological concentration) or with reliable analysis (e.g. with enough independent replicates) have been published. We therefore proposed that the effect of AT1R-activation on gene expression is related cross-activity with other pathways (e.g. EGFR) (discussed from line 586).
fig1a for 48h, why not the EAU group?
The bioinformatics analysis showed that most of the gene expression regulation is observed after 24h. Moreover, the qualitative synergistic effect induced by the combination of EGF and AngII observed after 24h was not after 48h. We decided not to perform an additional sequencing run to collect data for this condition after 48h, since there was virtually no effect of EGF and AngII at this time-point anymore.
What are the difference of the two figures in lower panel of fig2? It should be marked as a b c d for fig 2.
The left graph shows the log2 fold change distributions of the down-regulated genes, while the right one shows the log2 fold change distributions of the up-regulated genes. The figure and the related legend were modified based on these suggestions.
L397 Table 1, what the "regulators" mean? are they transcription factors? If not, how about the transcription factors changed in the stimulations? You may checked the AnimalTFDB database for TF reference.
We used here the “upstream regulator analysis” function of the QIAGEN Ingenuity Pathway Analysis (IPA) Software. This latter allows predicting potential regulators (transcription factors but also e.g. receptors, cytokines, enzymes…) of the observed changes in gene expression. These predictions are based on the IPA database, which contains information from curated literature. This procedure has the advantage to consider several types of molecules and is therefore not based on TF binding sites motifs.
The figure legend for fig 3 is unclear, especially for the fig 3b. what are the two figures? what are the lines with difference color? etc.
Similarly to the lower panel of Figure 2, the left and the right graphs of Figure 3b show the log2 fold change distribution for down- and up-regulated genes respectively. The figure and the related legend were also modified based on these suggestions.
So for the results and conclusion of this paper, which kind of genes are qualitative synergy or quantitative synergy. These should be mentioned in the results and abstract.
The genes that were quantitatively synergistically regulated and therefore showed a significant difference between their expected FC and measured FC are listed in Table 2 and Supplementary Fig S7. We primarily focused on two of them, Klf15 and Spp1, as they were previously described as involved in vascular integrity and we wanted to verify if the cross talks between the two receptors may influence such relevant genes. Further experimental validation were made for these two genes and the results are presented in Figure 6.
On the other hand, for the genes undergoing qualitative synergistic regulation, we did not select specific ones to study further but rather applied an untargeted approach (cf. Figure 4 and the GO term enrichment results). Because these results gave us hints that the qualitatively synergistically regulated genes were involved in cell cycle and DNA replication, we experimentally checked which impact had the simultaneous activation of EGFR and TP (BrdU incorporation and cell senescence).
Both the link between qualitative synergism and cell cycle/senescence, and the regulation of Klf15 and Spp1 are mentioned in the abstract and discussed later on.
Reviewer 2 Report
I like the attention and effort put into the visualization of the data, however, some letter directed organization of the figures would be nice (especially figure 2). I think some of the results (especially the result sub-headers) are worded a bit too strongly. For instance, I don't think you properly tested/proved there was simultaneous receptor activation or true synergistic effects.
Author Response
I like the attention and effort put into the visualization of the data, however, some letter directed organization of the figures would be nice (especially figure 2). I think some of the results (especially the result sub-headers) are worded a bit too strongly. For instance, I don't think you properly tested/proved there was simultaneous receptor activation or true synergistic effects.
The figures 2 and 3 and the related legends were adjusted.
We considered a response as “synergistic” when the effect induced by two stimuli is higher (or lower) than the simple addition of the effects of these two stimuli taken separately. Our results and analyses show that the simultaneous activations of more than one receptor have a synergistic effect on gene expression regulation since some genes required the combination of two stimuli to be actually regulated (what we defined here as qualitative synergism). Additionally, we could show that the expression of some genes was much more influenced by the combination of two stimuli than by a single one (called here “quantitative synergism”). The most striking example here was the gene Spp1 that was slightly regulated by EGFR-activation or TP-activation but showed a drastic increase when both stimuli where combined simultaneously (cf. Fig.6).